

# Incorporation of an invasive plant into a native insect herbivore food web

Menno Schilthuizen[1,2,3], Lúcia P. Santos Pimenta[2,4], Youri Lammers[1], Peter J. Steenbergen[2], Marco Flohil[5], Nils G.P. Beveridge[1,2], Pieter T. van Duijn[1,6], Marjolein M. Meulblok[1,6], Nils Sosef[1,6], Robin van de Ven[1,6], Ralf Werring[1,6], Kevin K. Beentjes[7], Kim Meijer[3], Rutger A. Vos[1,8], Klaas Vrieling[2], Barbara Gravendeel[1,2,6], Young Choi[2,9], Robert Verpoorte[2], Chris Smit[3] and Leo W. Beukeboom[3]

[1] Endless Forms group, Naturalis Biodiversity Center, Leiden, the Netherlands
[2] Institute for Biology Leiden, Leiden University, Leiden, the Netherlands
[3] Groningen Institute for Evolutionary Life Sciences, University of Groningen, Groningen, the Netherlands
[4] Departamento de Química, Instituto de Ciências Exatas, Universidade Federal de Minas Gerais, Belo Horizonte, Minas Gerais, Brazil
[5] ServiceXS, Leiden, the Netherlands
[6] University of Applied Sciences Leiden, Leiden, the Netherlands
[7] Biodiversity Discovery group, Naturalis Biodiversity Center, Leiden, the Netherlands
[8] IBED, University of Amsterdam, Amsterdam, the Netherlands
[9] Natural Products Laboratory, Leiden University, Leiden, the Netherlands

Corresponding author
Menno Schilthuizen,
menno.schilthuizen@naturalis.nl

## ABSTRACT

The integration of invasive species into native food webs represent multifarious dynamics of ecological and evolutionary processes. We document incorporation of *Prunus serotina* (black cherry) into native insect food webs. We find that *P. serotina* harbours a herbivore community less dense but more diverse than its native relative, *P. padus* (bird cherry), with similar proportions of specialists and generalists. While herbivory on *P. padus* remained stable over the past century, that on *P. serotina* gradually doubled. We show that *P. serotina* may have evolved changes in investment in cyanogenic glycosides compared with its native range. In the leaf beetle *Gonioctena quinquepunctata*, recently shifted from native *Sorbus aucuparia* to *P. serotina*, we find divergent host preferences on *Sorbus*- versus *Prunus*-derived populations, and weak host-specific differentiation among 380 individuals genotyped for 119 SNP loci. We conclude that evolutionary processes may generate a specialized herbivore community on an invasive plant, allowing prognoses of reduced invasiveness over time. On the basis of the results presented here, we would like to caution that manual control might have the adverse effect of a slowing down of processes of adaptation, and a delay in the decline of the invasive character of *P. serotina*.

## INTRODUCTION

The introduction and subsequent explosive spread of non-native species is seen as one of the main environmental disturbances threatening ecosystems globally (*Glowka, Burhenne-Guilmin & Synge, 1994*; *Gurevitch & Padilla, 2004*; *Simberloff, 2011*). Not all

introduced species will eventually successfully establish themselves and spread invasively (*Williamson & Fitter, 1996*). For example, populations of colonists may die out due to disease or adverse environmental conditions (*Rodriguez-Cabal, Williamson & Simberloff, 2013*). Nonetheless, the numbers of environmentally problematic exotics are increasing worldwide (*Butchart et al., 2010*). This even holds for parts of the world that are traditionally seen as sources, rather than recipients of exotic species, such as Europe (*Hulme et al., 2009*; *Van Kleunen et al., 2015*).

One potential explanation for the invasiveness of an introduced species is the so-called enemy release hypothesis, ERH (*Keane & Crawley, 2002*; *Liu & Stiling, 2006*), which states that, because the introduced species has not coevolved with the native biota, release from specialized parasites and predators causes explosive population growth.

Enemy release may cause the initial spread, but the subsequent population dynamics are more complex, and influenced by evolutionary processes. Reduced selection pressure for defences against specialist herbivores may result in the evolution of changed energy investment. For example, the plant may evolve stronger allocation of resources towards growth and reproduction and/or towards defence against generalists (*Blossey & Nötzold, 1995*; *Joshi & Vrieling, 2005*; *Zangerl & Berenbaum, 2005*; *Prentis et al., 2008*; *Whitney & Gabler, 2008*). However, at the same time, native herbivores may evolve the ability to locate and feed on introduced species (*Vellend et al., 2007*; *Pearse & Hipp, 2014*). Therefore, the course of the establishment of an introduced species is complex, with population dynamics modified by evolution: over time, the community of natural enemies attacking an introduced species tends to expand (*Brändle et al., 2008*) and the adverse impact of invasive species tends to wane (*Williamson, 1996*; *Simberloff & Gibbons, 2004*; *Blackburn, Lockwood & Cassey, 2009*; *Dostál et al., 2013*). This may be due to evolution in both the invader and the species it interacts with (*Vellend et al., 2007*). However, a species' invasive character is often considered static, and management policies rarely consider the possibility that it may change due to evolutionary adaptation (*Whitney & Gabler, 2008*).

One prominent invasive plant species in Europe is the black cherry, *Prunus serotina* Ehrh, native of eastern North America and considered a "forest pest" in Europe after widespread planting as auxiliary tree in pine plantations throughout the 20th century (*Schütz, 1988*; *Bakker, 1963*; *Starfinger et al., 2003*). Being bird-dispersed, it has been rapidly invading forested and open habitats (*Deckers et al., 2005*). In many European countries (*Starfinger et al., 2003*), it is now considered one of the most important threats to habitat quality of vegetation on dry, acidic, and/or poor soil, such as dunes and moorland (Fig. 1; *Godefroid et al., 2005*). In the Netherlands, for example, *P. serotina* has increased in distribution and abundance by at least two orders of magnitude during the second half of the 20th century (*Tamis et al., 2005*). Current control measures (chemical and mechanical eradication) are temporary and cosmetic (*Starfinger et al., 2003*). Nonetheless, they are costly: *Reinhardt et al. (2003)* conservatively calculated the annual cost of *P. serotina* control in Germany to be ca. 25 million euros.

Possibly the initial spread of *P. serotina* was facilitated by an absence of natural enemies; for example, *Reinhart et al. (2003)* found that, in the native range, soil pathogens inhibit the establishment of *P. serotina* seedlings near conspecifics, whereas in the invaded range,

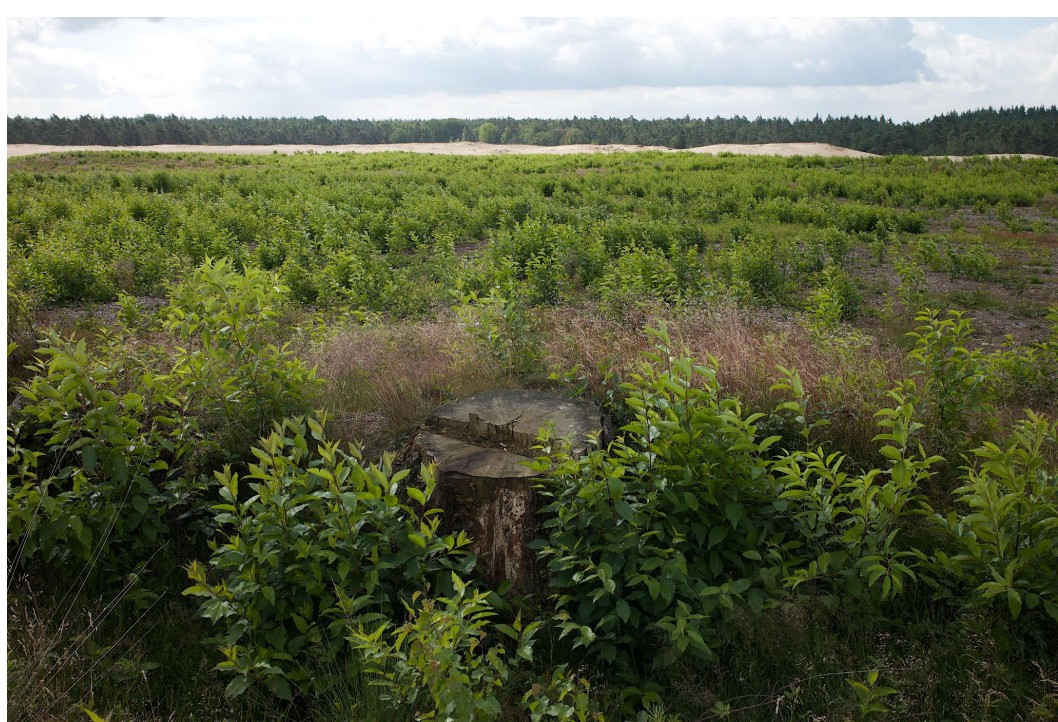

**Figure 1** **In open habitats, such as this moorland in the Netherlands, *Prunus serotina* may spread invasively, as this carpet of seedlings shows.** (photo credit: Ruud Lardinois, Stichting Kritisch Bosbeheer).

the species-specific soil community facilitates establishment. However, it is to be expected that the rich resource which *P. serotina* constitutes will provide adaptive opportunities for phytophagous insects to exploit. Such an evolutionary process will be even more likely if *P. serotina* represents an enemy-free space for herbivores (see *Feder (1995)* and *Karolewski et al. (2014)* for examples in other plants), and if it has been evolving reduced herbivore defences (*Blossey & Nötzold, 1995*). The changes in chemical defences may be complex. *Joshi & Vrieling (2005)* found that invasive plants may increase energetically cheap defences aimed at generalist herbivores, while reducing costly defences aimed at specialists when these specialists are no longer present.

Reports of native insects exploiting introduced *P. serotina* in Europe have been scarce throughout much of the 20th century, and have mostly concerned accidental feeding (by, e.g., moths, aphids, weevils, and leaf beetles; *Korringa, 1947*; *Hille Ris Lambers, 1971*; *Moraal, 1988*; *Klaiber, 1999*; *Fotopoulos, 2000*). Simultaneously, at least among nature management workers, a widespread belief has been maintained that the strong cyanogenic properties of the species, stronger than in *P. padus* (*Poulton, 1990*; *Swain, Li & Poulton, 1991*; *Santamour, 1998*; *Hu & Poulton, 1999*; *Fitzgerald, 2008*; *Pimenta et al., 2014*), have prevented native insect herbivores from colonizing it (*Nyssen et al., 2013*; *Anonymous, 2014*). More recently, however, studies from France, Germany, the Netherlands, and Poland are beginning to suggest that a community of native herbivores may in fact be accumulating on *P. serotina* (*Karolewski et al., 2014*; *Wimmer & Winkel, 2000*;

*Winkelman, 2005; Nowakowska & Halarewicz, 2006; Żmuda et al., 2008; Boucault, 2009; Halarewicz & Jackowski, 2011; Meijer et al., 2012; Karolewski et al., 2013*).

In this paper, we investigate the composition of the insect herbivore community feeding on *P. serotina* in the Netherlands. Because congenerics are likely to have been an important source of colonists, we compare the *P. serotina* herbivore community with the one occurring locally on *P. padus*, its closest native relative in the Netherlands (*Bortiri et al., 2001*). To obtain an impression of the accumulation of herbivory in *P. serotina*, herbivore damage in both *Prunus* species is quantified on the basis of herbarium records. We then investigate the impact that two conditions may have had on herbivore presence: cyanogenic defence compounds and parasitoid attack, in both *Prunus* species. Finally, as an example of the adaptive evolution that specialist *P. serotina* herbivores may have undergone, we studied host preference and genomics in one particular *P. serotina* herbivore, the leaf beetle *Gonioctena quinquepunctata*.

## MATERIALS AND METHODS

### Sampling herbivore communities on *P. serotina* and *P. padus*

The insect community feeding on both *Prunus* species was sampled in Nationaal Park Zuid-Kennemerland (52° 25′N, 4° 35′E), a partly forested area of coastal sand dunes near Haarlem, the Netherlands. Sampling was done by traversing a $2 \times 2$ km area in the old, forested dunes, and haphazardly selecting 300 individuals (150 of each species). We took care that on each day, roughly equal numbers of *P. padus* and *P. serotina* were investigated. Where possible, individuals of the two species were sampled in alternation. Sampling was done manually (no tools like nets, beating trays, or exhausters were used) in spring and early summer of 2009 (3 days), 2010 (10 days), and 2012 (8 days), by a single person inspecting, for 5 min., leaves, twigs, flowers, and fruits up to c. 2.5 m above ground level. All insects feeding or ovipositing on the host plant were stored in 96% ethanol. To obtain measurements on the actual amount of foliage searched, we replicated the above sampling method in September 2015 on 10 and 8 trees, respectively, of *P. serotina* and *P. padus*, and counted the numbers of leaves and lengths of twigs searched. We also determined fresh weights of ten leaves of each of the two plant species. Insects were identified morphologically, with help from experts (see 'Acknowledgements'). The 2009 and 2010 Geometridae and Tortricidae were identified by sequencing of the Cytochrome Oxidase I DNA-barcode region (e.g., *Van Nieukerken, Mutanen & Doorenweerd, 2011*) and the "animal identification" module in BOLD (www.boldsystems.org). All 2009 and 2010 specimens were deposited in the collections of Naturalis Biodiversity Center (container codes BE90711–90716). Because of improper curation, the specimens from the 2012 sampling were discarded after identification. We adopted *Leather*'s (*1985*) host range indicators of G (generalist, feeding on multiple plant families), R (feeding on Rosaceae only), P (on *Prunus* only), and M (monophagous, feeding on *P. padus* only). In addition, we categorized species that are specialized on non-Rosaceae (e.g., *Quercus*-specialists) as O ("other"). Differences in species richness for each of these categories were compared between both host species and tested for significance with a chi-square

test. Natuurmonumenten (Ruud Luntz) permitted us to work in Nationaal Park Zuid-Kennemerland under permit No. 19 of 2008. Dunea (Harrie van der Hagen) permitted us to work in Meijendel by permission 25/2/2013.

## Herbivory history on *Prunus padus* and *Prunus serotina*

We used historical accessions of *P. padus* and *P. serotina* in the herbarium collection of Naturalis Biodiversity Center to produce time-series of insect herbivory in the Netherlands for both hosts. Herbivory was assessed by a method of our own design, as a percentage of leaves on a herbarium specimen that showed pre-collection insect damage (post-collection damage by herbarium beetles was recognized and recorded, but not included in the herbivory data). We are aware of the fact that some botanists may preferentially have collected undamaged branches, so these estimates of herbivory are to be treated as conservative. We assessed changes of herbivory over time by Pearson tests on linear correlation coefficients.

## Parasitization of caterpillars

Within the same $2 \times 2$ km area as mentioned above, we sampled 173 and 110 live caterpillars from 43 *P. padus* and 32 *P. serotina* trees, respectively, between May 18th and June 3rd, 2011. All caterpillars were reared in individual vials. If a caterpillar metamorphosed into an adult moth or butterfly, it was considered unparasitized. If a parasitoid wasp or fly emerged, the host was considered parasitized. Caterpillars or pupae from which no adult insect had emerged by June 19th, were dissected in ethanol or Ringer's solution to determine the presence or absence of parasitoid eggs, larvae, pupae, or adults (*Zchori-Fein et al., 2001*). When found, these hosts were also considered as parasitized. Models describing the binominal response variable "parasitized" (Y/N) with combinations and interactions of the following explanatory variables: tree, method, xylosteana, and tree-ID (which was added as a random effect) were created and analysed in R 2.12.1 (*R Development Core Team, 2010*). "Tree" was the caterpillar's host plant species (*P. padus/P. serotina*). "Method" was the way a caterpillar was determined to have been parasitized or not (dissected in ethanol, dissected in Ringer's solution, or reared to adult or parasitoid emergence). "Xylosteana" indicated if the caterpillars belonged to the most commonly encountered species, *Archips xylosteana* (TRUE) or another species (FALSE). Of the identified caterpillars, all other species were not present in sufficient numbers (<8) for species-level analysis.

## Determination of cyanogenic glycosides

We analysed secondary plant compounds for 57 of the *P. padus* and 56 of the *P. serotina* plants for which we sampled herbivores in 2012 (see above). Immediately after each herbivore sampling, we harvested five young leaves and five old leaves from each tree, and kept these in separately labelled bags in a Dewar flask with solid $CO_2$ in the field. All samples were ground under liquid nitrogen and freeze-dried. We carried out NMR-analysis as described previously (*Pimenta et al., 2014*; *Kim et al., 2003*; *Kim, Choi & Verpoorte, 2010*). Briefly, extracts in $CH_3OH$-d4 and $KH_2PO_4$ buffer in $D_2O$ (1:1) were quantitatively analysed for prunasin and amygdalin, using 1H-NMR spectroscopy on a 500 MHz Bruker DMX-500 spectrometer (Bruker, Karlsruhe, Germany). Purity of quantitated 1H-NMR

signals was evaluated using several two-dimensional NMR experiments. Correlations were investigated between concentrations of each of the cyanogenic glycosides and herbivore load. We treated generalists (category G, see above) and specialists (categories R, P, M, and O) separately. In view of the high numbers of *Yponomeuta evonymellus* and *Rhopalosiphum padi* on some *P. padus* trees, we log-transformed the specialist herbivore load for *P. padus*. The relative amounts of cyanogenic glycosides were calculated per sample by taking the integrals in buckets $\delta 5.92$ (for prunasin) and $\delta 5.88 + \delta 5.84$ (for amygdalin). Correlations were tested with parametric Pearson's tests for the data on generalists and (in view of the large numbers of samples devoid of specialists) with non-parametric Spearman's tests for the data on specialists.

## A specialist herbivore's food preference for the original *Sorbus* vs. the novel *Prunus serotina*

We selected the oligophagous leaf beetle *G. quinquepunctata* for a case study of host preference. We chose this species because (i) it has very recently (probably in the early 1990s) colonized *P. serotina* in north-central Europe (*Klaiber, 1999*; *Halarewicz & Jackowski, 2011*; *Meijer et al., 2012*; *Mazderek et al., 2015*); (ii) it is a specialized species, originally feeding chiefly on rowan, *Sorbus aucuparia* (*Wimmer & Winkel, 2000*; *Koch, 1992*). Within a circle with 6-km radius around Eelde (53° 08′N, 6° 34′E), this beetle only feeds on the original native host *S. aucuparia* and the novel introduced *P. serotina* (not on any other hosts), and is equally abundant on both (*Meijer, 2013*). In May 2011, 83 adults and 138 larvae were collected from *S. aucuparia* and 63 adults and 57 larvae were collected from *P. serotina*, and kept separate by collection locality and host plant. These were used in host choice experiments: one *S. aucuparia* and one *P. serotina* branch (with 3–5 leaves each) was placed in a bottle filled with water, which was then placed in the centre of a 0.25 m$^3$ cage. Between one and five adults or between two and 10 larvae were selected from one of the live, host-specific collections and placed on the plug in the neck of the bottle. Each experiment was conducted with individuals from only one of the two hosts, and each individual was tested only once. Adults and larvae were not mixed within an experiment. After 21 h, the position for each individual was recorded and the animals were returned to their respective live collections. The test was performed 107 times. Tests were carried out on animals collected within a two-week period and were begun on the date that they were collected. We then tested for host preference using a GLM with binomial distribution. The model included the fixed factors of original host plant, life stage (larva or adult), interaction between original host plant and life stage, collection date, locality of origin, and cage (multiple cages were used). The effect of each factor was tested by removing one factor and comparing the complete model with the reduced model, and to do this successively with each of the factors, using ANOVA. Host preference in *G. quinquepunctata* was tested with a proportion test, by comparing the host choices for all animals, depending on their host of origin. All analyses were done in R (*R Development Core Team, 2010*).

## Genomic differentiation in host-specific subpopulations of a specialist herbivore

Using the same *G. quinquepunctata* specimens from Eelde as above, after finishing the host choice tests, we chose one adult individual from each host plant and obtained full genome sequences from these using paired-end forward-reverse sequencing on an Illumina HiSeq 2000. We pooled the data from both *G. quinquepunctata* sequencing runs and used this for a single *de novo* assembly. We assembled the data using Abyss (*Simpson et al., 2009*) with a *k*-mer length of 23 and a *k*-mer coverage of 3, values which we optimized using KmerGenie (*Chikhi & Medvedev, 2013*). We saved all produced contigs longer than 200 bp. We then mapped the data from both samples separately against these contigs using BWA (*Li et al., 2009*) at default settings and used Samtools (*Li & Durbin, 2009*) to call the SNPs in the BWA alignments. We looked up the SNP positions in the alignments for both samples and filtered based on the following criteria: the positions were both homozygous for different alleles between the samples, had a coverage of at least $10\times$ in each sample, had flanking regions that were at least 100 bp long with a minimum combined coverage of at least 15x with a maximum of 2 heterozygous positions. We identified the contigs containing valid SNP positions by BLASTing them against the GenBank nucleotide database and removing all non-arthropod contigs. Based on the remaining SNPs, we made a random selection of 128 SNPs (Table S5), all from different contigs, for which we designed primers using the Kraken software (LGCgenomics). Subsequently, in June 2014, again within the same 6-km radius around Eelde, we collected a new set of individuals from both hosts at five localities (Norg-1, Norg-2, Kleibos, Appelbergen, and Noordlaarderbos); 206 from *S. aucuparia*, and 173 from *P. serotina*. We performed DNA extractions on head + thorax using the NucleoMag 96 Tissue kit (Macherey-Nagel Gmbh & Co., Düren, Germany) on the KingFisher Flex magnetic particle processor (Thermo Scientific). DNA was diluted to 1 ng/μl and analysed in uniplex on the LGC Genomics SNP-genotyping line according to manufacturer's instructions. SNPs were detected using the KASP technique (*Semagn et al., 2014*). Genotypes were called using the Kraken software. We discarded five loci that did not yield scorable SNP-patterns and four loci that deviated from Hardy-Weinberg equilibrium, leaving 119 loci. Missing data were scattered over loci and samples and amounted to 2.9% of the total data set. We assessed population differentiation by Analysis of Molecular Variance (AMOVA), as well as by a Structure analysis (*Pritchard, Stephens & Donnelly, 2000*; *Excoffier & Lischer, 2010*). For Structure, standard settings were used and 10 replicates were performed for $K = 2$ to $K = 10$. The results were uploaded to Structure Harvester and a delta K plot was used to determine the number of groups (*Earl & VonHoldt, 2012*). We used a hierarchical AMOVA with host plants nested within localities, and we repeated the same AMOVA on a locus-by-locus basis.

## RESULTS

### Sampling herbivore communities on *Prunus serotina* and *Prunus padus*

Our sampling method covered on average, per tree, 258 (±136 s.d.) and 141 (±91 s.d.) leaves of *P. serotina* and *P. padus*, respectively. Given mean fresh weights of *P. serotina* and

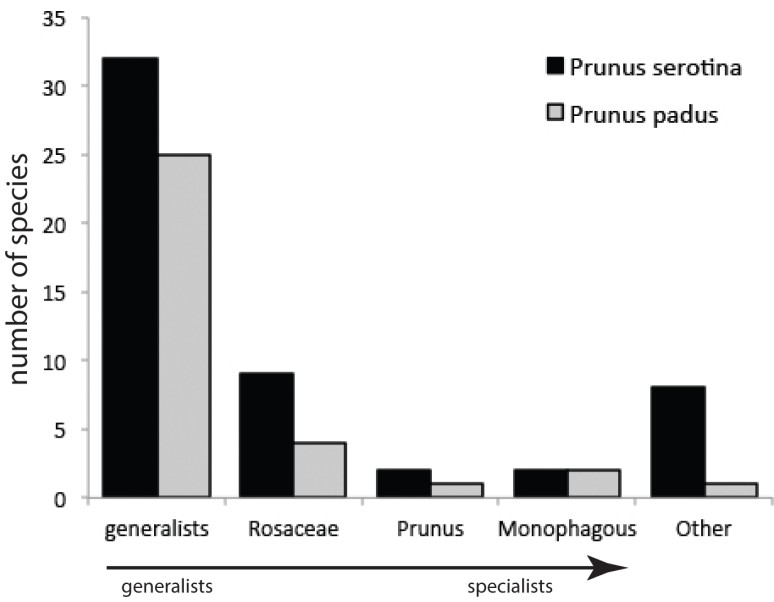

**Figure 2  Numbers of species from different categories of generalist and specialist insect herbivores sampled from *Prunus padus* and *Prunus serotina*.**

**Table 1  Data on insect herbivores communities sampled.**

|  | *N* (trees) | Leaf material searched | *N* (ins.) | *S* (ins.) | *N*[a] (ins.) |
|---|---|---|---|---|---|
| *P. serotina* | 150 | 17.025 kg | 794 | 64 | 748 |
| *P. padus* | 150 | 19.245 kg | 3,732 | 39 | 1,113 |
| Total | 300 | 36.270 kg | 4,523 | 72 | 1,860 |

**Notes.**
[a]Without *Yponomeuta* and *Rhopalosiphum*.
*N*, number of individuals ("load"); *S*, number of species.

*P. padus* leaves of 0.44 and 0.91 g, respectively, the amounts of foliage searched in 5 min were 113.5 g and 128.3 g for *P. serotina* and *P. padus*, respectively. After correction for the 1.13×more foliage searched in *P. padus*, we found that *P. serotina* harbors a 4.15-fold lower density but almost two-fold higher species diversity of herbivorous insects (Table 1; Table S1) than *P. padus*. The higher herbivore load on *P. padus* is, however, largely due to only two monophagous species, *Y. evonymella* (Lepidoptera: Yponomeutidae) and *R. padi* (Hemiptera: Aphididae), which usually occur in dense "nests" and "colonies", respectively (*Leather, 1985*). These two species were found on *P. serotina* at much lower densities and usually only as single individuals. Almost half of the herbivore specimens found on *P. padus* belong to these two species. We did not find a difference in the proportions of specialists versus generalists on the native and the non-native host (Fig. 2): both species carried similar (chi-square = 4.13; $P = 0.38$) proportions of each of the four categories of host range (G, generalists; R, Rosaceae-specialists; P, *Prunus*-specialists; M, *P. padus* monophages; and O, other—mostly *Quercus*—specialists).

### History of herbivory on *Prunus padus* and *Prunus serotina*

Herbarium records (Table S2) for *P. serotina* ($n = 96$; 2,817 leaves) showed a more than two-fold increase in herbivory (proportion damaged leaves) from 18.8% to 40.6% over the past 170 years ($r = 0.262$; $P = 0.0099$, $df = 94$; Pearson test; Fig. 3A). For *P. padus* ($n = 222$; 6,612 leaves), herbivory has remained stable at c. 35% over the past two centuries ($r = -0.020$; $P = 0.766$, Pearson test; Fig. 3B). In the most recent year (2013) we found no significant difference between the herbivory in *P. padus* (40%) and *P. serotina* (41%) ($T$-test; $P = 0.53$).

### Parasitization of caterpillars

The percentages of parasitized caterpillars on both *Prunus* species were not significantly different (*P. padus*: 55/173, 32%; *P. serotina*: 43/110, 39%; chi-square $= 1.58$; $P = 0.21$). Tables of explanatory variables and response variables are presented in Table S8. A third of all collected specimens belonged to *Archips xylosteana*. A test of independence of the explanatory variable tree explaining the response variable "parasitized" was not significant (chi-square $= 1.58$, $df = 1$, $P = 0.20$). A full generalized linear model was used to described the response variable "parasitized" as a three-way interaction between "tree", "method", and "xylosteana". The full model was not significant, and after simplifying the model by steps, the only explanatory variable to affect parasitization significantly was the method used to determine if a specimen was infected by a parasitoid ($P < 0.01$). The identified parasitoids mostly belonged to Ichneumonidae, Braconidae, and Tachinidae.

### Determination of cyanogenic glycosides

In the NMR-analyses (Table S3), we found that the concentration of cyanogenic glycosides (prunasin and amygdalin combined) per unit leaf dry weight is similar in both *Prunus* species. Mean concentrations in young and old leaves differed by <5% in each plant species. In both plant species, the ratio prunasin : amygdalin was c. 3 : 1. Generalist and specialist herbivores showed different relations with cyanogenic glycoside concentrations, and the responses to prunasin differed from those to amygdalin. Specifically, we found that the generalist herbivore load was not correlated with prunasin ($R = -0.08$, $P = 0.39$, both in *P. prunus* and *P. serotina*), but increased with amygdalin concentration ($R = 0.24$ and 0.36; $P = 0.01$ and 0.0001, respectively, in *P. padus* and *P. serotina*), whereas the specialist herbivore load increased with prunasin concentration, and decreased with amygdalin concentration, but significantly so only in *P. padus* (of which the amygdalin relationship would lose significance after Bonferroni correction; see statistical test results given in Fig. 4).

### A specialist herbivore's food preference for the original *Sorbus* vs. the novel *Prunus serotina*

At the end of the host choice experiment, 52% of all experimental *G. quinquepunctata* were present on one of the host plants. Individuals collected on *S. aucuparia* showed a significant preference for *S. aucuparia* (69.7 $\pm$ 3.1%) over *P. serotina* ($P < 0.0001$). However, individuals collected on *P. serotina* showed no significant preference for either host. Similar patterns were found in both adults and larvae: individuals from *S. aucuparia* preferred their original host (75.9 $\pm$ 7.0 % for adults, $P < 0.0001$, and 65.9 $\pm$ 9.0% for
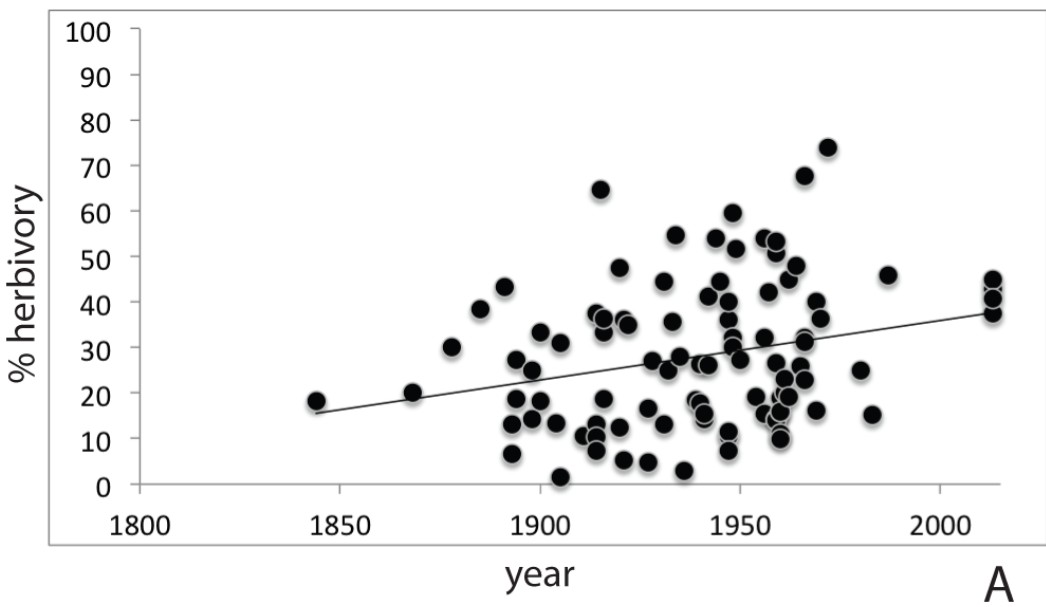

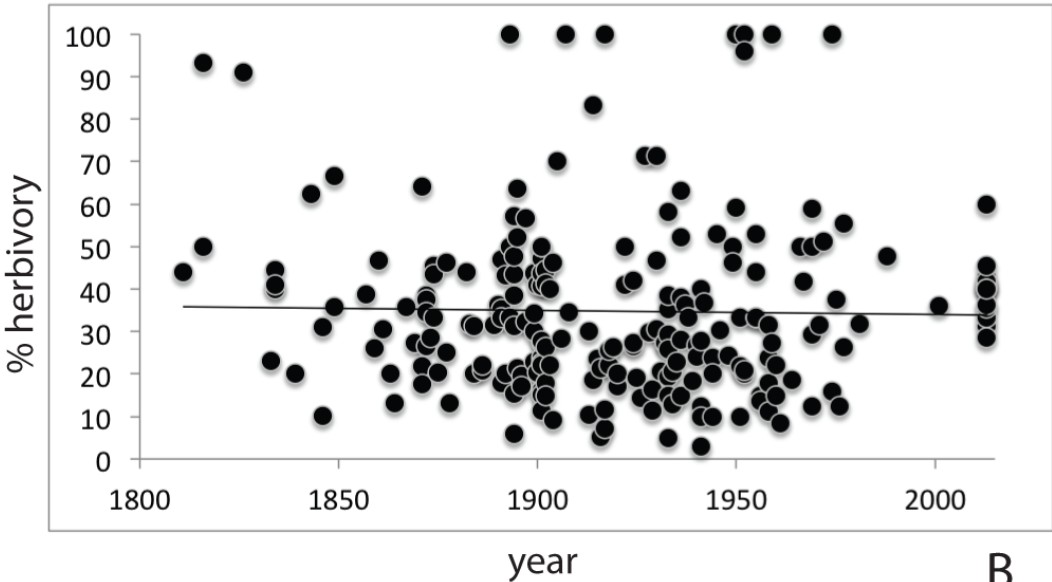

**Figure 3** Herbivory over time as derived from herbarium records; (A) *Prunus serotina*; (B) *Prunus padus*.

larvae, $P = 0.0003$); individuals from *P. serotina* showed no preference ($58.7 \pm 9.1\%$ for adults, $P = 0.2077$, and $57.9 \pm 14.3\%$ for larvae, $P = 0.2893$). Full test results are available in Table S7.

## Genomic differentiation in host-specific subpopulations of a specialist herbivore

Illumina sequencing of a *G. quinquepunctata* larva from *S. aucuparia* gave 157,327,896 reads, and 191,340,606 reads were obtained from an adult beetle found on *P. serotina*. The

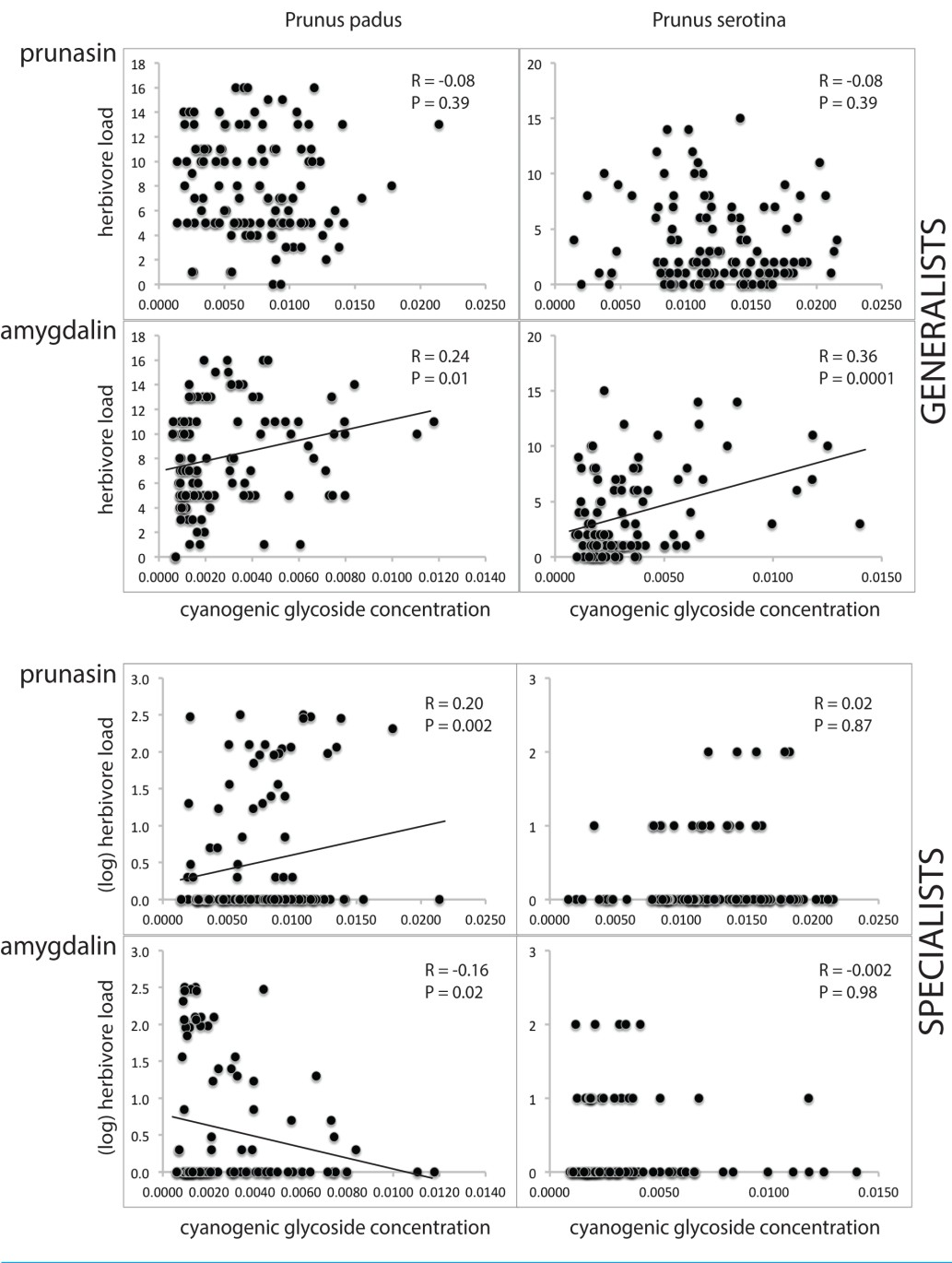

**Figure 4 Cyanogenic glycosides and herbivory.** *Prunus padus* is shown in the left column, *Prunus serotina* in the right column. Data for generalist herbivores are shown in the top four graphs (separately for prunasin and amygdalin), and for specialist herbivores in the bottom four graphs (also separately for prunasin and amygdalin). Pearson correlation coefficients (for the data for generalists) and Spearman's rho (for the data for specialists) and corresponding *P*-values are given, and regression lines are shown for significant relationships. Note that the *P*-value for amygdalin vs. specialists in *P. padus* does not remain significant after Bonferroni correction. Herbivore loads (on the *y*-axis) are given as counts of individuals per tree, except in the case of specialists on *P. padus*, where the log was taken. Cyanogenic glycoside amounts (on the *x*-axis) are given as NMR signal integrals.

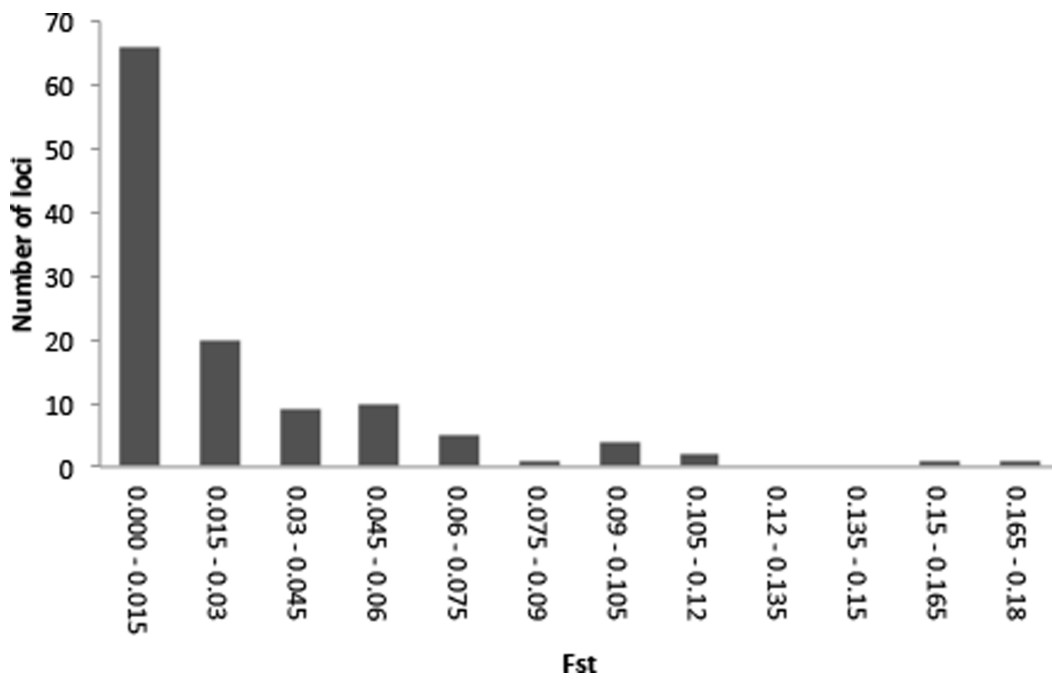

**Figure 5** Frequency distribution of per-locus pairwise (*Prunus–Sorbus*) $F_{ST}$ values for *Gonioctena quinquepunctata*.

*de-novo* assembly with Abyss resulted in 438,237 contigs longer than 200 bp. The data were deposited in the NCBI short read archive under BioProject accession code: PRJNA277307. A total of 729 usable SNPs were obtained from the SNP discovery. To assess genetic differentiation in both host-specific subpopulations, we genotyped 379 individuals from both hosts at each of five localities, for the selected 119 SNP loci (Table S4). Our Structure analysis (Text S1) failed to detect overall genetic differentiation between the populations on both host plants: the highest posterior probability was for $K = 2$, but these two groups did not correspond with host plant nor with locality. The hierarchical AMOVA with host plants nested within localities, showed significant ($P < 0.01$) differentiation between host plants in each locality. On a locus-by-locus basis, the AMOVA revealed 13 loci that were significantly differentiated between the two subpopulations from *P. serotina* and *S. aucuparia*, two of which remained significant after strict Bonferroni correction (Table S6). The distribution of per-locus pairwise (*Prunus–Sorbus*) $F_{ST}$ values (Fig. 5) also shows that at least two loci are outliers. Homology searches in Genbank for these SNP loci yielded no matches with genes of known function.

## DISCUSSION

Our inventories show that the invasive *P. serotina* in the Netherlands harbours a surprisingly rich community of herbivores. Although the densities were lower than on native *P. padus*, the species diversity was greater. Also, contrary to expectations, the *P. serotina* herbivore community contained similar proportions of specialists versus generalists as the one on *P. padus*. The only species strikingly absent from *P. serotina* were two abundant *P. padus*

monophages, *Y. evonymellus* and *R. padi*. Consistent with *Leather (1985)*, both species were responsible for more than two thirds of all insects found feeding on *P. padus*, whereas they occurred on *P. serotina* only in small numbers (we found only a single *Y. evonymellus* caterpillar and a single *R. padi* colony on *P. serotina*). Nonetheless, laboratory data (*Kooi, Van de Water & Herrebout, 1991*) and field data from Poland (*Karolewski et al., 2014*) suggest that at least *Y. evonymellus* has the potential to feed on *P. serotina*. *Karolewski et al. (2014)* state that in Poland, the latter species has progressed from avoiding *P. serotina* altogether to feeding and developing on it massively over the past decade. The near-absence from *P. serotina* of this herbivore in our study area suggests that a similar colonization event may not yet have taken place, but this may change in the near future, possibly aided by long-distance gene flow from the populations in Poland. Another striking difference between both hosts is the relatively large numbers of non-Rosaceae specialists on *P. serotina*. While some of these may be accidental "tourists", the high number of individuals for some of these species (e.g., the *Quercus*-specialist *Harpocera thoracica*) is noteworthy.

These results add to a body of data on insect herbivory on native versus non-native plants (reviewed in, e.g., *Liu & Stiling, 2006*; *Colautti et al., 2004*; *Meijer, 2013*). Although these studies tend to show that introduced plant species, especially those with powerful chemical defences, are poor hosts for native herbivores, exceptions have also been found of introduced species hosting a larger number of species than closely-related native plants (*Novotny et al., 2003*). The rich herbivore community on non-native *P. serotina*, and especially the high number of specialist species, fits with the observation that the food web supported by a non-native plant expands as time since initial introduction increases (*Brändle et al., 2008*). Although *P. serotina* was introduced into Europe earlier (*Schütz, 1988*), it only became common in the 20th century (*Starfinger et al., 2003*). Its increasing abundance in Dutch ecosystems over the past 80 years may have been the phase during which most of the herbivore community has built up. Indeed, while our study of leaf damage in herbarium specimens cannot reveal the diversity of herbivores, it does show that herbivore damage, and therefore perhaps herbivore load, has gradually doubled over this period, while that on *P. padus* has not changed. Today, at least based on our herbarium records, herbivory levels in both plant species appear to be similar (despite the lower herbivore load that we found in our inventory for *P. serotina*—see above).

In theory, the rapid assembly of this community may have been aided by the presence of an enemy-free space for the insect herbivores. If local parasitoids, for example, are not adapted to using *P. serotina* volatiles as a cue for attraction to a possible patch in which to find hosts, this may have helped the establishment of herbivore populations on the introduced plant (*Feder, 1995*; *Harvey & Fortuna, 2012*). Indeed, *Karolewski et al. (2014)* found reduced parasitization of one species, *Y. evonymella* on *P. serotina*. However, we find that current attack rates of caterpillars by parasitoids do not differ between *P. padus* and *P. serotina*.

After an initial period of reduced specialist herbivory in the non-native range, *P. serotina* may have shifted its investment in chemical defences in favour of those aimed at generalists (*Joshi & Vrieling, 2005*). Cyanogenic glycosides are generally considered to be systemic, non-inducible, and energetically cheap chemical defences aimed primarily at generalist

herbivores (*Gleadow & Møller, 2014*). However, our phytochemical data suggest that, in *P. padus* (and, less clearly, in *P. serotina*), the Rosaceae-specific compound amygdalin has a positive relationship with generalist load but a negative one with specialist load, whereas the more widespread compound prunasin has a positive correlation with specialist herbivore load, while lacking any clear relation with generalist load. It would be tempting to compare the levels and ratios of prunasin and amygdalin in today's *P. serotina* populations in the Netherlands with those reported for the native American population. However, we only have access to a single American study (*Santamour, 1998*), which, moreover, employed somewhat different methods (see below), so we do so with considerable hesitation. *Santamour (1998)* reported a summertime HCN production in native American *P. serotina* corresponding to 29.6 mg cyanogenic glycosides per g fresh leaf material (see Text S2). In an earlier study of 22 Dutch *P. serotina* trees (*Pimenta et al., 2014*), we found on average 30.4 mg cyanogenic glycosides per mg *dry* leaf material. As *P. serotina* dry leaf weight is 36% of fresh leaf weight (see Text S2), this might suggest that total cyanogenic glycoside content in the invaded range could be about two- to threefold lower than in North America. Also, Santamour found prunasin : amygdalin proportions of 22:1, whereas we found a ratio of 3:1. In the Dutch *P. serotina*, prunasin investment might therefore have decreased, with amygdalin content remaining more or less constant. Since both the absolute and relative amounts of prunasin and amygdalin content have a genetic basis (*Santamour, 1998*), these results might indicate that cyanogenic glycoside defence has, after the introduction into Europe, adapted to the novel herbivore communities. With a mean age at first reproduction of only 5.2 years (*Deckers et al., 2005*) and evidence, in general, of rapid evolution of defence in invasive plants (*Felker-Quinn, Schweitzer & Bailey, 2013*), such a quick evolutionary change is not implausible. However, since *Santamour (1998)*, *Pimenta et al. (2014)* and the present study appear to be the only available quantifications of prunasin and amygdalin in *P. serotina*, and since the range of phenotypic plasticity in cyanogenic glycoside content is unknown, more data, with more comparable methods, are needed before this conclusion can be substantiated. Moreover, we stress that our results and their discussion refer only to the cyanogenic potential (HCNp), whereas the true defence potential is a combination of HCNp and HCNc, cyanogenic capacity, which is a function of glucosidase presence and activity. Since the latter is unknown in this study, we implicitly assume that HCNp is an indicator for cyanogenic defence, which may only be partly true and is known to differ between specialists and generalists (*Ballhorn, Kautz & Lieberei, 2010a*).

The accumulation of the herbivore community on *P. serotina* may also have involved evolutionary processes within the insect community itself. One possibility is that all present-day herbivores were able to feed and reproduce on *P. serotina* from the moment the new host was introduced. However, this would not explain the *slow* increase in herbivory that our herbarium data show: highly mobile insects with short generation times would have established on the new host instantaneously, rather than gradually. It is therefore likely that adaptive evolution in the herbivores played an important role in the assembly of this community over time.

As a possible example of this scenario, we performed a case study on one specialist herbivore, the leaf beetle *G. quinquepunctata*, which has recently colonized *P. serotina* from its original host, rowan (*S. aucuparia*). We find indications of weak differentiation in host preference and SNP-loci on *Sorbus*- versus *Prunus*-derived beetle individuals. We found that individuals collected on *Sorbus* retained a significant host preference for this host, whereas beetles collected from *Prunus* showed no preference for *Prunus* over *Sorbus*. We found the same host preference in adults and larvae, although presumably host choice is made mostly in the mobile, adult stage. While these results do not necessarily imply genetic differentiation, as learning may be involved as well (*Salloum, Colson & Marion-Poll, 2011*), our SNP-analysis does show indications of weak genetic differentiation, with several loci showing divergence, and potentially linked to regions that are under disruptive, host-imposed selection. In other words, the introduced *P. serotina* may have selected for weak, incipient divergence (*Vellend et al., 2007*; *Nosil & Feder, 2011*) in this particular herbivore. Whether such selection will allow further sympatric speciation, in this herbivore or others, depends not only on the different selection regimes imposed by the different host plants, but also on the mount of gene-flow between the populations feeding on the two hosts (*Nosil & Feder, 2011*).

Overall, our results indicate that, since its introduction, a rich and diverse herbivore community has accumulated on *P. serotina*. It is possible that evolutionary adaptations in these herbivores as well as in the plant itself have played an important role in shaping this community. Adaptation may have involved niche widening in generalist herbivores, incipient genetic divergence in specialists, as well as adjustments of chemical defences in the host plant.

These results may have implications for invasive species management. It may be expected that the gradual evolutionary integration of a novel plant species in a native herbivore food web may eventually reduce its invasive character to the point where it attains the status of non-harmful, naturalized neophyte. Whether this will happen in the case of *P. serotina* depends on a number of factors. In this paper, we dealt with herbivorous insects only, whereas plant demographics are affected by a much broader spectrum of natural enemies. *Reinhart et al. (2003)* and *Van der Putten (2000)* suggested that its invasiveness may be more due to an absence of belowground interactions (with the *Prunus*-pathogenic fungus *Pythium*, for example) than aboveground interactions. However, preliminary studies in the Netherlands indicate the presence of local *Pythium* populations that are powerful in attacking introduced *P. serotina* (Tamis & Van der Klugt, pers. comm., 2015). Furthermore, *Ballhorn, Pietrowski & Lieberei (2010b)* and *Ballhorn (2011)* found that in cyanogenic plants a trade-off exists between defence against herbivores and against fungal pathogens, which is an additional complication not yet considered. A final point of concern is the intensity of the regime of natural selection. Presently, manual control of mature *P. serotina* in many European habitats is reducing the continued exposure of the host to its potential herbivores. On the basis of the results presented here, we would like to caution that this might have the adverse effect of a consequent slowing down of processes of adaptation, and a delay in the decline of the invasive character of *P. serotina*.

## ACKNOWLEDGEMENTS

The following experts helped with insect identifications: Theodoor Heijerman (Coleoptera: Curculionoidea), Willem Ellis (leaf miners), Marja van der Straten (Lepidoptera), Erik van Nieukerken and Camiel Doorenweerd (Lepidoptera), Ping-Ping Chen (Hemiptera), and Kees van Achterberg (Hymenoptera). Ruud Luntz (Natuurmonumenten) and Hubert Kivit (PWN) provided important details on *P. serotina* distribution in Zuid-Kennemerland. Luc Willemse, Kees Koops, René Glas, Kees van den Berg, Jekaterina Tkacova, Daniel Cisneros Torres, Renda Remmerswaal, Esther van der Meer, and Bertie-Joan van Heuven helped in the lab. Leni Duistermaat, Wil Tamis, and Rinny Kooi provided details on *P. serotina* and its herbivores. Rienk Apperloo, Sjoerd Hobma, Anne Posthumus, and Marlijn Sterenborg helped with the experiments on *Gonioctena* host preference. Rick de Graaf, Stephen Pieterman, and Carla Stegehuis assisted in the *de novo* assembly. The photograph in Fig. 1 was kindly provided by Ruud Lardinois of Stichting Kritisch Bosbeheer, Dieren, the Netherlands.

### Funding

This study was financially supported by grants from the Uyttenboogaart-Eliasen Foundation and the Team for Invasive Exotics of the former Netherlands Ministry for Agriculture, Nature, and Food Quality. The funders had no role in study design, data collection and analysis, decision to publish, or preparation of the manuscript.

### Grant Disclosures

The following grant information was disclosed by the authors:
Uyttenboogaart-Eliasen Foundation.

### Competing Interests

Marco Flohil is an employee of ServiceXS, a company providing DNA services such as reported in this paper.

### Author Contributions

- Menno Schilthuizen conceived and designed the experiments, performed the experiments, analyzed the data, wrote the paper, prepared figures and/or tables.
- Lúcia P. Santos Pimenta and Youri Lammers performed the experiments, analyzed the data.
- Peter J. Steenbergen, Marco Flohil, Nils G.P. Beveridge, Pieter T. van Duijn, Marjolein M. Meulblok, Nils Sosef, Robin van de Ven, Ralf Werring and Kevin K. Beentjes performed the experiments.
- Kim Meijer conceived and designed the experiments, performed the experiments, analyzed the data, prepared figures and/or tables.
- Rutger A. Vos conceived and designed the experiments, analyzed the data.
- Klaas Vrieling analyzed the data.

- Barbara Gravendeel, Young Choi, Robert Verpoorte, Chris Smit and Leo W. Beukeboom conceived and designed the experiments.

## Field Study Permissions

The following information was supplied relating to field study approvals (i.e., approving body and any reference numbers):

Natuurmonumenten (Ruud Luntz) permitted us to work in Nationaal Park Zuid-Kennemerland under permit No. 19 of 2008. Dunea (Harrie van der Hagen) permitted us to work in Meijendel by permission 25/2/2013.

## DNA Deposition

The following information was supplied regarding the deposition of DNA sequences:

The data were deposited in the NCBI short read archive under BioProject accession code: PRJNA277307.

## Data Availability

The raw data has been supplied as Supplemental Information.

## Supplemental Information

Supplemental information for this article can be found online at http://dx.doi.org/10.7717/peerj.1954#supplemental-information.

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
