# Peer review of "Incorporation of an invasive plant into a native insect herbivore food web"

_PeerJ, doi:10.7717/peerj.1954_

## Round 0.1 · original submission · Minor Revisions

I agree with both reviewers found the study interesting, multi-faceted and overall well done. However, both of reviewers also had reservations about how several results were interpreted. Both reviewers had issues with how the cyanogenic glycoside data were interpreted, albeit for different reasons. This suggests to me that there may be a number of possible alternative explanations, and this section should probably be toned down. Reviewer 1 also took issue with the incorporation of museums specimens. I agree that it is problematic to compare modern-day estimates of herbivore damage to historical levels, due to sampling bias issues. There may be other explanations for the observed SNP patterns, so more caution is in order. Please respond to these criticisms, and other minor points raised by the reviewers.

·

Basic reporting

The manuscript “Incorporation of an invasive plant into a native insect herbivore food web” by Schilthuizen and co-worked investigates a time question: the role of invasive species in natural food webs. The paper is well written and easy to follow and overall I would love to see it published. However, I have a couple of major and some minor concerns – or suggestions for improvement.
Major concerns:
1. The way museum specimens are incorporated in this study is highly questionable. In general, using such specimens is an excellent idea but it does not work here very well. One of the issues (which the authors admit, line 385) is that plant specimens with high probability are biased towards low herbivore damage. Such specimens are “more pretty” and are more likely to be collected. Also, and that is for sure, the sampling method applied in this study was different than for the museum specimens. Thus, any statistics and subsequent statements are not valid. This whole passage either needs to be deleted or drastically rephrased. I tend towards deletion. Meaningful statistical analyses are NOT possible.
2. The potential role of cyanogenic glycosides needs to be reconsidered. Essential information regarding the cyanogenic features of the analyzed Prunus species is missing. The cyanogenic potential (HCNp) – the authors report here – is only one part of the story. The activity of β-glucosidases breaking down these precursors and critically determining the release of hydrogen cyanide per time (HCNc) is the other part of the story – which is completely neglected here. Generalist and specialist herbivores respond very differently to the HCNp and HCNc. Also variation in the efficiency of cyanogenesis in the interaction with herbivores with different feeding styles needs to be considered. I realize, that it is impossible at this stage to analyze the samples for β-glucosidase activity and HCNc but this needs to be discussed thoroughly (please see Ballhorn et al. (2010). Comparing responses of generalist and specialist herbivores to various cyanogenic plant features. Entomol Exp Appl 134:245-259.
3. Related to this issue: highly cyanogenic plants are frequently less defended against fungal pathogens than low(er) cyanogenic conspecifics. This aspect is missing as well. It would have been interesting to simultaneously quantify the occurrence of fungal lesion or to sequence plant samples with fungal specific primers. Again, this is hard to accomplish at this point but needs to be discussed (see for example Ballhorn et al. (2010). Direct trade-off between cyanogenesis and resistance to a fungal pathogen in lima bean (Phaseolus lunatus L.). J Ecol 98:226-236 and Ballhorn (2011). Constraints of simultaneous resistance to a fungal pathogen and an insect herbivore in lima bean (Phaseolus lunatus L.). J Chem Ecol 37:141-144.

Minor comments:
- Please check for consistence in terms of American vs. British English (harbor vs. harbor etc.)
- In the abstract it might be useful to mention the common names of the plant species studied as these might be more familiar to parts of the readership.

Experimental design

The use of museum specimens is problematic for this type of analysis. Further, method for determining herbivory/herbivore presence needs a reference!! Important!

Validity of the findings

see above; The conclusion derived from the comparative analyses of museum specimens and field collected leaves is not appropriate.

Additional comments

Good paper! But needs some more work.

Reviewer 2 ·

Basic reporting

All fine, well-written with appropriate and up-to-date literature citations

Experimental design

No problems with design, methods well-described.

Validity of the findings

Findings are valid, with a few caveats on the interpretation detailed below in comments to the authors.

Additional comments

This manuscript presents a study that thoroughly and systematically documents the herbivorous insect community that has accumulated on Prunus serotina in its invasive range in the Netherlands. The study documents the accumulation of natural enemies on P. serotina since its initial introduction, and explores potential mechanisms behind this phenomenon, such as changes in plant defenses and evolutionary change in insect populations. Overall I feel that the study was well-conducted and I do not see any major methodological problems. There are a few minor areas in which I would like to see a bit more nuance in the discussion.

First of all, I feel that the presentation of the relationship between generalist and specialist insect load and cyanogenic glycoside concentrations needs to be interpreted with caution. As it stands, the manuscript implies that the observed herbivore loads are a result of chemical defense concentrations. This is implied by language such as “herbivores responded differently to c.g. concentrations” and “herbivore load was not affected by prunasin.” (L168-175). This comes up again in the discussion through references to compounds attracting and deterring generalist vs. specialist herbivores (L264-267). While this might be the case, I think that the manuscript should avoid implication of cause an effect here, since herbivore loads can also influence plant defenses through induction. Given the design of the study, I’m not convinced that the dataset allows definitive identification of casue and effect.

I also felt that the authors should exercise caution in their interpretation of SNP data. Loci showing divergence may not necessarily be under disruptive selection (L309). Spatially separated populations could be drifting. I would re-word to ‘potentially liked to regions under disruptive selection’ rather than ‘presumably.’ I would also cut back the discussion of sympatric speciation to one or two sentences, as this is a bit of a stretch given the results of the study at hand.

A few other line by line comments:

L112: uneccessary repeat of ‘adaptive evolution’
L229: may have not yet (not ‘has’)
L231-234: Were the Quercus specialists observed consuming P serotina? Or just happened to land on them? What is the relative density of Quercus to P serotina in these forests?
L316: gene flow between populations feeding on two hosts, not between the hosts themselves
L344-346: I think this is an important point that needs to be recognized more in invasion management literature.

---

## Round 0.2 · accepted · Accept

The authors did an excellent job addressing the reviewers' criticisms. I was pretty sure that the comments addressed Reviewer 1's specific comments, but checked with him to confirm my opinion just in case. I think that all issues have been resolved, and the manuscript can be published.

·

Basic reporting

I still think that the use of museum specimens and resulting statistics are problematic for several reasons (stated in my initial review). However, as such a use of specimens in general is a really good idea and as the authors seem to be aware of the limitations of this approach...well, I guess I am fine with that.

I also really appreciate the author's efforts to incorporate the HCNp/HCNc issue as well as potential cyanide-mediated variation in pathogen resistance. Well done.

Experimental design

fine

Validity of the findings

Given the more tentative discussion of the presented findings I do not see any further issues with their validity.

Additional comments

Dear Authors,

I didn't mean to be too harsh with my criticism of the use of herbarium specimens. In fact, I feel this is an excellent use of such specimens. I agree that there is eventually no issue of an internal comparison. However, from my experience with herbarium specimens (including voucher specimens collected by myself) there is no doubt about a bias for for more 'pretty' samples - which makes sense as these show all the morphological characteristics required for species identification etc.

Anyway, it might be different with you samples. Again, good paper. I enjoyed reading it!

Daniel